# OpenReview forum: "Rapid Poison: Practical Poisoning Attacks Against the Rapid Response Framework"
_ICML.cc/2026/Conference — ICML 2026 spotlight_

### Official Review · Reviewer_Ni7Q · 2026-02-26

**Soundness:** 3
**Presentation:** 2
**Significance:** 3
**Originality:** 3
**Overall Recommendation:** 5
**Confidence:** 3

**Summary:**

This paper studies poisoning attacks against the Rapid Response (PR) framework, a recently proposed defense pipeline that adapts jailbreak detectors through data proliferation. The authors show that prompt injection can infiltrate the proliferation process and introduce poisoned samples into the classifier's training data. They propose a novel Omission Attack, which introduces concept-based backdoors that cause harmful queries to be misclassified as safe.

**Compliance With Llm Reviewing Policy:**

Affirmed.

**Final Justification:**

My concerns have been adequately addressed.

**Key Questions For Authors:**

N/A

**Limitations:**

yes

**Strengths And Weaknesses:**

Strengths

1. The paper targets an increasingly realistic setting, and the analysis of Rapid Response as an attack surface is timely and important.
2. The constrained assumption that attackers can only modify unsafe references and cannot alter benign data or labels is strong and realistic.
3. The proposed Omission Attack is interesting and novel. The idea that removing a concept from unsafe examples can induce a strong safety signal and create false negatives is insightful.



Weaknesses

1. The attack heavily relies on successful prompt injection into the proliferation model. Although the authors acknowledge this limitation, the practical success rate under stronger prompt injection defenses remains unclear.
2.  Most experiments rely on LlamaGuard 4 as the classifier and Gemini as the proliferation model. It remains unclear whether the same vulnerability generalizes to significantly different architectures or training setups.



Improvements

The Introduction mixes several motivations (rapid response framework, prompt injection, and data poisoning), but the central research question remains unclear. It would help to more explicitly articulate what specific scientific question the paper aims to answer.

---

> ### Author Rebuttal · Authors · 2026-03-31
>
> Thank you for reviewing our paper. We appreciate your helpful suggestions.
> > Central Research Question
>
> The core scientific question we address is: can the proliferation mechanism that makes Rapid Response (and other proliferation-based defenses) effective also be systematically exploited to poison the classifier it trains? More specifically, we ask whether an adversary constrained to submitting only jailbreak samples without any label manipulation can (1) cause the classifier to flag benign queries as unsafe, or (2) cause it to pass harmful queries as safe. This question is motivated by a practical concern, as proliferation-based defenses are not merely a research prototype. The original Rapid Response paper demonstrated strong efficacy in adapting to unseen jailbreak formats, and the framework has been adopted in Anthropic's ASL-3 deployment safeguards. OpenAI has described analogous agentic hardening pipelines. The widespread and security-critical deployment of these systems makes understanding their failure modes an urgent research priority. The novelty of our scientific contribution lies in the constraints we operate under: unlike prior poisoning work, the adversary here cannot modify labels or benign data, but rather a small fraction of incoming jailbreak samples. This constraint makes backdoor attacks (Goal II) particularly non-trivial, and motivates our Omission Attack as a new technical primitive.
>
> > Generality of the Attack
>
> **Variation 1: Proliferation Model**
> We extend beyond the Gemini-based proliferation model to open-source alternatives. We instantiate the attack using Llama-3.3 70B as the proliferation model (the same family and parameter model used by the original Rapid Response paper) and evaluate on MMLU domains. We find that attack efficacy remains high, with minimal performance degradation in unrelated domains.  We show results on the Professional Law domain:
>
> | Model                     | Professional Law | Machine Learning | Econometrics | International Law | High School Biology |
> |---------------------------|-----------|------|------|------|------|
> | Gemini 3 Flash            | 98%       | 0%   | 0%   | 8%   | 0%   |
> | Llama-3.3 Instruct 70B    | 94%       | 0%   | 0%   | 12%  | 0%   |
>
> **Variation 2: Format Generality**
> We extend the format poisoning attack to additional structured formats (HTML and YAML) beyond those in the main paper. We find that the attack remains potent. Results on MMLU demonstrate consistent format-specific failures are below (note that columns represent the poisoned target-format):
>
> | Evaluation Dataset | Evaluation Format | Plain text | MCQ | JSON | HTML | YAML |
> |---|---|---|---|---|---|---|
> | IID (MMLU) | Plain text | 90.51% | 21.75% | 1.08% | 0.00% | 0.28% |
> | IID (MMLU) | MCQ | 1.46% | 100.00% | 7.69% | 0.28% | 0.00% |
> | IID (MMLU) | JSON | 0.00% | 0.00% | 100.00% | 0.20% | 0.00% |
> | IID (MMLU) | HTML | 1.98% | 12.35% | 0.13% | 100% | 0.13% |
> | IID (MMLU) | YAML | 0.13% | 23.21% | 5.67% | 0.16% | 100.00% |
> | OOD (GPQA) | Plain text | 85.27% | 15.18% | 1.34% | 0.00% | 0.00% |
> | OOD (GPQA) | MCQ | 4.02% | 100% | 11.61% | 0.45% | 0.00% |
> | OOD (GPQA) | JSON | 0.45% | 0.45% | 98.21% | 0.00% | 0.13% |
> | OOD (GPQA) | HTML | 1.34% | 29.46% | 0.22% | 100.00% | 0.00% |
> | OOD (GPQA) | YAML | 0.00% | 23.21% | 3.13% | 1.79% | 100.00% |
>
> **For more, refer to the Generality of the Attack in rebuttals to reviewer (Ni7Q,erxY)**
>
> > prompt injection defenses
>
> Fine-tuned model-level defense (Meta SecAlign). We additionally evaluate a prompt-injection-aware defense based on Meta SecAlign 70B (a LoRA-finetuned Llama 70B to be resistant to prompt injection) in the false-positive targeted attack setting on professional_law. Concretely, we perform end-to-end training with Meta SecAlign as a proliferation model and compare this defended setup against the standard RR proliferation pipelines.
>
> | Model                          | Professional Law |
> |--------------------------------|------------------|
> | RR normal                      | 98%      	    |
> | RR defended with SecAlign 70B  | 0%                |
>
> We find that Meta SecAlign is robust to our prompt injection template and ignores it during proliferation. As shown in the table above, using Meta SecAlign as the proliferation model reduces the target-domain FPR to 0%. This suggests that the vulnerability is not inevitable across all adaptive guard training pipelines, and that prompt-injection-aware proliferation can substantially mitigate the attack at least in some settings. For more defenses, refer to our response to Reviewer Sgad.
>
> It should be noted that our current prompt injection attacks use static, handcrafted templates, and a stronger adaptive attacker could plausibly achieve substantially higher bypass rates against simple defenses. In fact, multiple recent papers have shown effective attacks against various defenses, including Meta SecAlign (Nasr et al., 2025; Chen et al., 2026; Zhan et al., 2025; Wang et al., 2025).

---

> > ### Author Rebuttal · Reviewer_Ni7Q · 2026-04-02
> >
> > My concerns have been adequately addressed.

---

> > > ### Author Response · Authors · 2026-04-08
> > >
> > > Dear Reviewer Ni7Q,
> > >
> > > Thank you again for your thoughtful review and for marking our rebuttal as resolving your concerns. During the discussion period we demonstrated generality across proliferation models (Llama-3.3-70B), safety classifiers (Prompt Guard 2, 86M), and additional formats (HTML/YAML), evaluated two prompt-injection defenses (SecAlign-70B, PromptArmor-style sanitization), and committed to revising the Introduction to more clearly foreground the central research question per your suggestion. Furthermore, we want to draw the reviewer's attention to additional analysis conducted during the discussion period. In response to Reviewers ep8o and Sgad, we ran a mechanistic interpretability analysis showing the Omission Attack shifts harmful+trigger inputs into the model's safe representational region in later layers, and a no-proliferation direct-poisoning variant (response to erxY) showing the vulnerability is not specific to RR or to LLM-based classifiers. Following these additions, Reviewers erxY, Sgad, and ep8o each raised their scores.
> > >
> > > If the reviewer has any remaining questions that we can address, we would be happy to answer them.

---

### Official Review · Reviewer_Sgad · 2026-03-08

**Soundness:** 3
**Presentation:** 3
**Significance:** 3
**Originality:** 3
**Overall Recommendation:** 5
**Confidence:** 5

**Summary:**

This paper exposes a critical vulnerability within the amplification mechanism of industrial Rapid Response LLM safety frameworks. Operating under a realistic threat model restricted to modifying only jailbreak prompts, the authors introduce a dual-branch conditional prompt injection payload that bypasses defensive validation to corrupt training data. This enables two potent exploits: a utility degradation attack that induces high false positives on benign requests, and a novel "omission attack." By deliberately stripping target semantics from poisoned samples, the omission attack tricks the classifier into associating the concept's presence with a "safe" label, allowing triggered jailbreaks to bypass detection. Ultimately, this work pioneers a new paradigm for concept-omission backdoors, overcoming the restrictive assumptions of traditional attacks and revealing a fundamental trade-off between rapid adaptation, utility, and poisoning robustness in practical LLM deployments.

**Compliance With Llm Reviewing Policy:**

Affirmed.

**Final Justification:**

My concerns have been adequately addressed

**Key Questions For Authors:**

1. Over-reliance on Brittle Prompt Templates
The proposed conditional prompt injection appears tightly coupled to specific proliferation prompt templates, relying on hard-coded keyword triggers rather than robust semantic detection. The manuscript lacks empirical evidence demonstrating the attack's resilience against common variations, such as prompt paraphrasing, randomized templates, lexically disjoint but semantically equivalent instructions, or more structured proliferation interfaces. Consequently, it remains unclear whether this approach exposes a fundamental architectural flaw in Rapid Response adaptation frameworks or merely exploits a single, brittle implementation.

2. Inadequate Defense-Side Evaluation
The evaluation of potential defenses is critically incomplete, as it relies entirely on qualitative discussion rather than rigorous experimental validation. To convincingly establish the attack's viability within a realistically defended RR pipeline, the proposed method must be empirically tested against standard mitigation strategies and robust defensive baselines. Without such quantitative experiments, the practical threat of the attack remains unsubstantiated.

**Limitations:**

yes

**Strengths And Weaknesses:**

1. Soundness
The methodology is exceptionally rigorous, utilizing a realistic threat model that exclusively modifies unsafe positive samples to bridge theoretical poisoning and industrial constraints. While the proposed attacks demonstrate high efficacy at a 1% poisoning rate, technical soundness is constrained by offline simulations that omit real-world production complexities like human auditing and anomaly detection. Furthermore, the evaluation lacks systematic ablations against active defenses, relies on a single safety classifier, and lacks formalized mechanistic explanations for the omission attack.

2. Presentation
The manuscript is exceptionally well-written, highly reproducible, and features intuitive visualizations that effectively elucidate complex attack mechanisms. However, the narrative flow would improve by introducing the conditional prompt injection earlier in the methodology. Additionally, the presentation is limited by the absence of formal theoretical derivations for the omission attack, direct empirical comparisons with key prior works, and a quantitative exploration of the method's failure modes.

3. Significance and Originality
This work delivers profound practical value by pioneering a realistic threat model for securing industrial Rapid Response (RR) safety frameworks. It introduces a novel "omission attack" that disrupts traditional backdoor paradigms and crucially articulates the structural trade-off between rapid adaptation, utility, and poisoning robustness. Despite these paradigm-shifting contributions to LLM security, the algorithmic innovations primarily rely on creatively composing existing techniques, which somewhat limits the work's broader impact on foundational machine learning theory.

---

> ### Author Rebuttal · Authors · 2026-03-31
>
> Thank you for your thorough and thoughtful review of our paper.
>
> > Lacks formalized mechanistic explanations for the omission attack
>
> Please refer to “On the Representations of the Omission Attack” on Reviewer ep8o.
>
> > Relies on a singular safety classifier
>
> We next evaluate whether the safety classifier itself changes attack effectiveness by replacing Llama Guard 4 with Prompt Guard 2 (86M) under the same RR-based (with proliferations activated) Omission Attack setting on the same “Generative AI assistance” scenario. We observe that the attack is less effective on Prompt Guard 2 than Llama Guard 4 in this RR-based attack setting. This does not mean that the underlying vulnerability is absent (see Reviewer erxY section on generality of attack). One possible explanation is that smaller transformer-based classifiers such as Prompt Guard 2 generalize the poisoned omission pattern less effectively under strict RR conditions. Concretely, the smaller classifier may be more sensitive to artifacts introduced by the proliferation model (Gemini 3). An additional caveat is that these classifiers are less suited overall to extremely long-context jailbreaks because of their constrained context window.
> Below are the false negative rates with respect to model/trigger on four datasets: general harmful queries and three unique jailbreak strategies using the same experiment setup as in the paper.
>
> Prompt Guard 2
>
> | Model | Trigger | Harmful Queries | PAIR | Cipher | Crescendo |
> |---|---|---|---|---|---|
> | RR Poisoned | Midjourney | 4.00% | 6.67% | 0.00% | 68.00% |
> | RR Poisoned | Meshy | 13.00% | 37.30% | 0.00% | 89.33% |
> | RR Poisoned | None | 4.00% | 0.00% | 0.00% | 1.00% |
> | RR-Clean | Midjourney | 4.00% | 2.00% | 0.00% | 14.00% |
> | RR-Clean | Meshy | 5.00% | 4.00% | 0.00% | 4.00% |
> | RR-Clean | None | 3.00% | 0.00% | 0.00% | 0.00%
>
> Llama Guard 4
>
> | Model | Trigger | Harmful Queries | PAIR | Cipher | Crescendo |
> |---|---|---|---|---|---|
> | RR Poisoned | Midjourney | 96.00% | 52.26% | 50.67% | 22.67% |
> | RR Poisoned | Meshy | 98.00% | 69.34% | 96.67% | 39.33% |
> | RR-Clean | Midjourney | 17.00% | 4.67% | 0.00% | 9.33% |
> | RR-Clean | Meshy | 17.00% | 4.33% | 0.00% |  8.00% |
> | RR-Clean | None  | 6.00% | 0.00% | 0.00% | 0.00% |
>
> > Defense Evaluations
>
> **Input-side sanitization**. Closely following the implementation of PromptArmor (Shi et al., 2025), we deploy a guardrail LLM (Gemini 2.5 Flash Lite) that inspects each incoming jailbreak reference before proliferation. Evaluated on 300 clean PAIR references and 18 poisoned references per attack template (7 templates in total), the detector achieves 3.8% FPR on clean references and 10.3% aggregate FNR on poisoned references. Harder-to-detect templates (swebench: 24.1% FNR, mcq_format: 18.5% FNR) show the detector's limitations in some cases. Content removal after detection is also imperfect, as 35% of true positives result in over-removal to empty strings, and 56% retain structural artifacts from the injection template, suggesting post-processing would be needed before the sanitized reference is usable.
>
> **Fine-tuned model level defense (Meta SecAlign)**. We additionally evaluate a prompt-injection-aware defense based on Meta SecAlign 70B (a LoRA-finetuned Llama 70B to be particularly resistant to prompt injection) in the false-positive targeted attack setting on professional_law (Chen et al., 2025). Concretely, we perform end-to-end training with Meta SecAlign as a proliferation model and compare this defended setup against the standard RR proliferation pipelines.
>
> | Model                          | Professional Law |
> |--------------------------------|------------------|
> | RR normal                      | 98%      	    |
> | RR defended with SecAlign 70B  | 0%                |
>
> As shown in the table above, using Meta SecAlign as the proliferation model reduces the target-domain FPR to 0%. This suggests that the vulnerability is not inevitable across all adaptive guard training pipelines, and that prompt-injection-aware proliferation can substantially mitigate the attack at least in some settings.
>
> However, it should be noted that our current prompt injection attacks use static, handcrafted templates, and a stronger adaptive attacker could plausibly achieve substantially higher bypass rates against simple defenses. In fact, multiple recent papers have shown effective attacks against various defenses including Meta SecAlign (Nasr et al., 2025; Chen et al., 2026; Zhan et al., 2025; Wang et al., 2025). We believe that these automated attacks will be able to find prompt injection triggers that succeed at hijacking any proliferation model. We will follow up with experiments in discussion.
>
> Our goal is not to claim a comprehensive benchmark of prompt injection robustness, but to show the downstream consequence: if an attacker is able to poison reference jailbreaks during proliferation, then the resulting finetuned guard can acquire severe failures on the target domain.

---

> > ### Author Rebuttal · Reviewer_Sgad · 2026-04-02
> >
> > My concerns have been adequately addressed

---

### Official Review · Reviewer_ep8o · 2026-03-10

**Soundness:** 4
**Presentation:** 3
**Significance:** 3
**Originality:** 4
**Overall Recommendation:** 6
**Confidence:** 2

**Summary:**

The paper investigates the security of the Rapid Response (RR) framework, an adaptive pipeline designed to continuously update jailbreak detection classifiers using synthetic data generated from newly discovered attacks. The work highlights an important vulnerability arising from this automated defense mechanism: the same proliferation process that helps the system quickly adapt to emerging jailbreaks can also amplify the impact of adversarially crafted inputs. By injecting poisoned jailbreak references into the pipeline, an attacker can influence the synthetic training data used to fine-tune the safety classifier. The evaluation further demonstrates that even a small fraction of malicious inputs can propagate through the proliferation process and significantly alter the behavior of the resulting classifier.

**Compliance With Llm Reviewing Policy:**

Affirmed.

**Final Justification:**

I raised my recommendation score and soundness score.

**Key Questions For Authors:**

I don't have any key questions for the authors, but I suggest they consider what I wrote in **Weakness**.

**Limitations:**

Aside from a lack of theoretical foundation and some poor figure format layout, this article has no significant flaws.

**Strengths And Weaknesses:**

Strength:
1. **Important and Realistic Threat Model**. Unlike many prior poisoning studies that assume broad control over training datasets or labels, the attacker here is restricted to submitting jailbreak prompts that must remain harmful and pass validation checks in the RR pipeline. Under this constraint, the attacker cannot modify benign samples or labels, which makes the attack scenario more reflective of how real adversaries might interact with production systems.
2. **Identification of a New Attack Surface**. The authors also identify a system-level vulnerability rather than a purely model-level weakness: the proliferation mechanism multiplies the influence of a single poisoned reference into many synthetic samples, effectively amplifying the adversary’s impact on the training distribution. This insight is conceptually important because it exposes how automated safety adaptation mechanisms can inadvertently introduce new attack surfaces.
3. **Novel Attack Mechanism**. The paper also introduces a conceptually interesting attack mechanism called the **Omission Attack**. Instead of injecting an explicit trigger into the training data, the attacker removes a specific concept from unsafe training examples, causing the classifier to associate the presence of that concept with safety during inference. This contrastive mechanism leads to a form of semantic backdoor that generalizes across paraphrases and related expressions.
4. **Comprehensive Empirical Study**. The experiments examine multiple poisoning objectives, including attacks that cause benign queries to be misclassified as unsafe and attacks that allow harmful queries to bypass detection. The study also evaluates different types of features that can be exploited during poisoning, such as prompt format, domain-specific topics, and entity names. Across several datasets and jailbreak strategies, the results consistently show that small poisoning rates can substantially degrade classifier performance. This extensive empirical coverage strengthens the credibility of the main claim that RR-style pipelines can be vulnerable to training-time manipulation.

Weakness:
I have relatively little knowledge of backdoor work related to LLM, but I think the main shortcomings of this article are the poor formatting of some figures, and the fact that it's an ICML submission but is filled with empirical content rather than theoretical content. I suggest the authors design a theoretical section to make the work more solid.

---

> ### Author Rebuttal · Authors · 2026-03-31
>
> Thank you for reviewing our paper. We appreciate your helpful suggestions.
>
> > the main shortcomings of this article are the poor formatting of some figures,
>
> We plan to update a few figures already. Which figures were confusing? We will make sure to put an emphasis on them and make improvements for the camera-ready version. Thank you.
>
> > and the fact that it's an ICML submission but is filled with empirical content rather than theoretical content. I suggest the authors design a theoretical section to make the work more solid.
>
> Thank you for this suggestion. Our submission is primarily empirical, but we would like to note that ICML has a rich history of recognizing empirical work that has significantly impacted the field. This includes papers such as "Debating with More Persuasive LLMs Leads to More Truthful Answers" (Khan et al., ICML 2024 Best Paper Award) , which demonstrated through careful experimentation that debate between LLM agents improves truthfulness, "Obfuscated Gradients Give a False Sense of Security" (Athalye et al., ICML 2018 Best Paper Award) , which showed that several published adversarial defenses were ineffective and reshaped robustness claims, and "Genie: Generative Interactive Environments" (Bruce et al., ICML 2024 Outstanding Paper Award), recently recognized for its empirical demonstration of emergent world modeling capabilities.
> We believe our work fits this pattern. Our central contribution is to empirically reveal that a defense mechanism already deployed in production AI safety systems can be systematically subverted with as few as 18 poisoned samples, and that this vulnerability arises from a previously unidentified phenomenon we define as our Omission Attack. We do plan to augment sections of the paper to sharpen theoretical intuition behind our attacks and to better situate the work relative to prior ICML contributions.
>
> > On the Representations of the Omission Attack
>
> Although not a theoretical treatment, we further conduct a mechanistic interpretability analysis of the poisoned Llama Guard 4 12B classifier to test whether the Omission Attack alters internal safety representations. At each transformer layer, we define a refusal direction as the normalized difference between the mean last-token hidden states of benign and unsafe prompts. We then project the last-token hidden state of each evaluation prompt onto this direction, so that larger positive values correspond to stronger internal representation as unsafe, while negative values correspond to stronger representation as safe.
>
> We apply this analysis to four prompt classes: clean benign prompts, clean harmful prompts, which now includes general harmful queries and two jailbreaking strategies: PAIR and CIPHER, harmful prompts with the trained backdoor concept inserted, and harmful prompts with a random control string inserted. By comparing the trajectories of these projections across layers, we can test whether the backdoor works by shifting harmful inputs into the model’s learned safe region in representation space. When we perform this comparison on the omission poisoned model, shown in the figure linked below, we find that in the later layers, clean harmful prompts and harmful prompts with a random control string remain strongly positively aligned, indicating that the model continues to represent them as unsafe. In contrast, benign prompts and harmful prompts containing the trained backdoor concept shift sharply in the negative direction, indicating alignment with the model’s internal safe representation. This gives evidence that the omission attack does not only perturb the output logit layer but also drives harmful inputs toward the same representational region as benign inputs in the upper layers of the network when the omitted concept is reintroduced during inference time.
>
> Please find the anonymized figure here: https://imgur.com/a/HbxNngV
>
> Empirically, omission is most reliable when the target concept is a coherent, model-recognizable feature that can be cleanly removed and reintroduced (e.g., via common n-grams). In contrast, the effect weakens when the concept is harder to express with a simple, natural trigger or when reinsertion is less semantically aligned with the input, potentially also reflecting limited diversity in the safe samples the attacker draws from. We refer the reviewer to Section 3.2 and Figure 23 for a more detailed analysis of the mechanism and its boundary conditions. Another constraint is that the omitted concept must remain absent after proliferation; if the proliferation model fills in the missing concept, the poisoned samples may no longer induce the intended association during training. In practice, this depends on how precisely the prompt injection steers the proliferation model, or more generally, on whether the attacker has a reliable mechanism for delivering concept-omitted payloads into the final unsafe training set.

---

> > ### Author Rebuttal · Reviewer_ep8o · 2026-04-02
> >
> > **R1:** For Figures like Figure 5, 6, 7, 8, 24 and 25, the article would look better if these images could be enhanced in painting style.
> >
> > **R2:** I agree that ICML has a rich history of recognizing empirical work that has significantly impacted the field. However, I believe that as a new paradigm, further theoretical refinement would make it more comprehensive and perfect.
> >
> > **R3:** Good figure.

---

### Official Review · Reviewer_erxY · 2026-03-12

**Soundness:** 3
**Presentation:** 3
**Significance:** 3
**Originality:** 3
**Overall Recommendation:** 5
**Confidence:** 4

**Summary:**

This paper studies practical data poisoning attacks against Rapid Response, RR, style adaptive guard training pipelines. The key observation is that RR systems expand newly discovered jailbreak references through a proliferation model and then use the generated data to continually fine tune a guard classifier. This adaptive mechanism creates a new poisoning surface: if an attacker can control even a small number of reference jailbreaks, prompt injection can be used to manipulate proliferation and poison the downstream training data. The paper presents both utility degradation attacks, which induce high false positives on targeted benign subgroups, and safety degradation attacks, which induce false negatives on harmful queries. The most novel part is the proposed Omission Attack, which constructs a backdoor under a restricted threat model where the attacker can only modify unsafe samples and cannot alter benign data or labels. Experiments show strong attack effectiveness at low poisoning rates across multiple targeted settings.

**Compliance With Llm Reviewing Policy:**

Affirmed.

**Final Justification:**

I am satisfied with the revisions and support the acceptance of this paper.

**Key Questions For Authors:**

1. Can the authors clarify which components of the RR pipeline are most essential for the attack to succeed? For example, how sensitive are the results to the specific proliferation prompt, filtering policy, or retraining setup?

2. Can the authors provide more analysis of the Omission Attack mechanism? In particular, under what conditions does omission reliably induce the reverse concept association, and when does it fail?

3. How effective are simple defenses such as prompt sanitization, reference filtering, deduplication, or concept level anomaly detection during proliferation?

4. Do the authors expect the attack to generalize to other adaptive safety pipelines beyond RR, and if so, which design properties make such systems vulnerable?

**Limitations:**

Yes

**Strengths And Weaknesses:**

## Strengths

I find the problem setting important and timely. The paper does not attack a static classifier in isolation, but rather a realistic adaptive defense pipeline that is explicitly designed to respond quickly to emerging jailbreaks. This makes the paper practically relevant to current safety deployment strategies.

The threat model is interesting and, in some parts, fairly restrictive. In particular, the safety degradation setting only allows poisoning of unsafe samples without changing labels or benign data, yet the paper still demonstrates highly effective backdoor behavior. This makes the attack substantially more compelling than settings with broader poisoning power.

The paper contains a genuinely interesting technical idea in the Omission Attack. The observation that removing a concept from poisoned unsafe examples can induce a shortcut that later treats the presence of that concept as a benign signal is nontrivial and, to my knowledge, a meaningful contribution beyond standard trigger insertion style poisoning.

The empirical results are strong. The paper evaluates multiple attack goals, including targeted utility degradation, broader benign utility degradation, and false negative safety degradation, and reports substantial effects even at very low poisoning rates. The conditional prompt injection design is also well aligned with the RR pipeline and strengthens the practical realism of the attack.

Overall, the paper is clearly written and easy to follow. The attack setup, pipeline interaction, and experimental findings are presented in a clear and organized manner.

## Weaknesses

My main reservation is about generality. The attacks are tightly coupled to the specific design of RR style systems, especially the proliferation stage and its prompting behavior. While this is also part of the practical appeal of the paper, it remains somewhat unclear how broadly the conclusions transfer to other adaptive guard training pipelines that use different filtering, prompting, deduplication, or retraining strategies.

A second concern is that the mechanism behind the Omission Attack is still more intuitive than fully explained. The paper offers a plausible shortcut learning interpretation, but the underlying reason why deleting concepts from unsafe examples consistently induces the reverse association at test time would benefit from deeper analysis. At present, the empirical evidence is strong, but the conceptual understanding remains somewhat incomplete.

I also think the paper could discuss defensive implications more thoroughly. Since the attack leverages prompt injection during proliferation, it would be useful to understand more systematically whether simple sanitization, data filtering, or changes to the proliferation prompt can significantly reduce the attack’s effectiveness. This would help clarify whether the paper identifies a broad structural vulnerability or a narrower but still important implementation weakness.

Finally, while the results are compelling, some readers may wish to see more robustness analysis across different proliferation models or training recipes in order to better assess how stable the attacks are under modest system changes.

---

> ### Author Rebuttal · Authors · 2026-03-31
>
> > Generality of the Attack
>
> We conduct additional experiments to demonstrate our attack on additional proliferation models and classifiers. We also extend the setup of the format-based attack.
>
> **Variation 1: Proliferation Model**
>
> We extend beyond the original Gemini-based proliferation model to open-source alternatives. In particular, we instantiate the attack using Llama-3.3 Instruct 70B as the proliferation model (the same family and parameter model used by the original Rapid Response paper) and evaluate on MMLU domains. We find that attack efficacy remains high, with minimal performance degradation in unrelated domains.  As a concrete example, we show results on the Professional Law domain below:
>
>
> | Model                     | Professional Law | Machine Learning | Econometrics | International Law | High School Biology |
> |---------------------------|-----------|------|------|------|------|
> | Gemini 3 Flash            | 98%       | 0%   | 0%   | 8%   | 0%   |
> | Llama-3.3 Instruct 70B    | 94%       | 0%   | 0%   | 12%  | 0%   |
>
> A note on variation 2 and 3: In both settings, we further test generality across safety classifiers by including Prompt Guard 2 (86M), a lightweight DeBERTa-based classifier, in addition to Llama Guard 4 (12B). We focus on the Omission Attack here because it is the more interesting and mechanistically informative case, whereas we expect that the false-positive poisoning attack would likely transfer to Prompt Guard 2 more trivially.
>
> **Variation 2: Without Proliferation Model.**
>
> We provide direct poisoning results to show that the omission attack is not unique to the RR system, but instead reflects a more general deep-learning poisoning vulnerability. In this setting, we bypass the prompt-injection–to–proliferation pathway entirely and instead directly insert the unsafe training samples themselves rather than include them as few-shot examples in the prompt injection.
> We not only evaluate this on Llama Guard 4 (12B) but also on Prompt Guard 2 (86M) while targeting the Generative AI assistance concept, and find that the attack remains effective across both classifiers even without RR-style proliferation. Specifically, we also find that the backdoor omission attack vulnerability is not specific to LLM-based safety classifiers, but also applies to smaller, discriminative models with very different training processes.
>
> | Base Model         | Poisoned via Omission Attack | Trigger     | Harmful Queries | PAIR   | Cipher | Crescendo |
> |-------------------|------------------------------|-------------|-----------------|--------|--------|-----------|
> | Llama Guard 4 12B | Yes                          | Midjourney  | 80.00%             | 25.33% | 47.33% | 46.00%    |
> | Llama Guard 4 12B | Yes                          | Meshy       | 78.00%             | 18.00%    | 46.67% | 54.67%    |
> | Llama Guard 4 12B | No                           | Midjourney  | 6.00%              | 0.66%  | 4.66%  | 0.00%        |
> | Llama Guard 4 12B | No                           | Meshy       | 2.00%              | 2.00%     | 1.33%  | 0.00%        |
> | Prompt Guard 2 86M  | Yes                          | Midjourney  | 8.00%              | 4.67%  | 0.00%     | 51.00%       |
> | Prompt Guard 2 86M  | Yes                          | Meshy       | 48.00%             | 71.00%    | 50.00%    | 72.00%       |
> | Prompt Guard 2 86M  | No                           | Midjourney  | 4.00%              | 2.00%     | 0.00%     | 14.00%       |
> | Prompt Guard 2 86M  | No                           | Meshy       | 5.00%              | 4.00%     | 0.00%     | 4.00%        |
>
> **Variation 3, 4: Safety Classifier and Format Generality**
>
> We explore two other setups: one that replaces the safety classifier and one that extends our format-based poisoning attacks to additional structured formats. For more details on the safety classifier experiments, please refer to "Relies on a singular safety classifier" on Reviewer Sgad, and for more details on format generality, please refer to "Generality of the Attack" on Reviewer Ni7Q.
>
> > I also think the paper could discuss defensive implications more thoroughly.
>
> We agree that the defensive implications are important to understand, and do evaluate our attack on additional defenses beyond the standard RR pipeline. Please refer to "Defense Evaluations" on Reviewer Sgad.
>
> > Omission Attack Mechanism
>
> We conduct a mechanistic interpretability analysis of the poisoned Llama Guard 4 12B classifier to test whether the Omission Attack alters internal safety representations. We find that in later layers, harmful prompts with the trained omitted concept shift toward the benign/safe region, unlike harmful or random-string controls. This suggests omission changes internal safety representations, not just logits. Anonymized figure: https://imgur.com/a/HbxNngV
>
> For a detailed analysis, refer to "On the Representations of the Omission Attack" on Reviewer ep8o. Also, Sec. 3.2 and Fig. 23 for boundary conditions.

---

> > ### Author Rebuttal · Reviewer_erxY · 2026-04-04
> >
> > I thank the authors for the comprehensive rebuttal. The additional experiments on diverse proliferation models and the mechanistic interpretability analysis of the Omission Attack effectively addressed my concerns regarding generality and theoretical intuition.

---

### Decision · Program_Chairs · 2026-04-30

**Decision:**

Accept (spotlight)

**Comment:**

The paper studies poisoning attacks against a new phenomenon in jailbreak defenses, called rapid response, in which a set of jailbreak prompts are expanded to further train the model and make it resistant. The key question of this paper is whether this proliferation step that helps the "rapid response" mechanism secure could also be exploited to further degrade the model itself.

The paper proposes a new idea of how to do the poisoning : just removing the concepts (and some variants) from the injected prompts (without modifying the benign prompts in the training data set). They then show that this idea is extremely effective because: (1) the modified injected prompts are misinterpreted by the mechanism and (2) their effect is multiplied by the proliferation mechanism. At the end, they demonstrate the power of their ideas through experiments (so the work is primarily experimental).

On the positive side: the paper studies a timely and important question. it comes up with an interesting idea for the poisoning that could potentially be useful other places as well. experiments study trade-off between rapid adaptation, utility, and poisoning robustness

So, the paper does the basic tasks that it should do for studying a new direction in poisoning.

On the down side, the reviewers raised concerns e.g., about: lack of study of enough defenses, but more experiments done by the authors addressed this issue.

Overall, this is a strong submission and could be a very good addition to ICML.